# Wind Field Digital Twins Sandbox System for Transmission Towers

**DOI:** 10.3390/s23218657

**Published:** 2023-10-24

**Authors:** Chenshuo Zhang, Yunpeng Li, Chun Feng, Yiming Zhang

**Affiliations:** 1School of Civil and Transportation Engineering, Hebei University of Technology, Tianjin 300401, China; cs_zhang1997@163.com (C.Z.); 202311601008@stu.hebut.edu.cn (Y.L.); 2Institute of Mechanics, Chinese Academy of Sciences, Beijing 100190, China; fengchun@imech.ac.cn

**Keywords:** digital twin, transmission tower, wind load, CDEM, numerical simulation

## Abstract

Given the digitalization trends within the field of engineering, we propose a practical approach to engineering digitization. This method is established based on a physical sandbox model, camera equipment and simulation technology. We propose an image processing modeling method to establish high-precision continuous mathematical models of transmission towers. The calculation of the wind field is realized by using wind speed calculations, a load-wind-direction-time algorithm and the Continuum-Discontinuum Element Method (CDEM). The sensitivity analysis of displacement- and acceleration-controlled transmission tower loads under two different wind direction conditions is conducted. The results show that the digital model exhibits a proportional relationship with the physical dimensions of the transmission tower model. The error between the numerical simulation results and the experimental results falls within a reasonable range. Nodes at higher positions of the transmission tower experience significantly higher forces compared to those at lower positions, and the structural forms with larger windward projected areas yield similar simulation results. The proposed digital twin system can help monitor the performance of structural bodies and assess the disaster degree in extreme conditions. It can guide specific maintenance and repair tasks.

## 1. Introduction

The reliability and safety of power grid infrastructure have emerged as pivotal factors impacting both the country’s energy security and economic development, making power grid engineering a cornerstone industry within the national economy. Currently, the inadequate levels of informatization, delayed data collection and a lack of intelligent decision-support tools have constrained the progress and quality of power grid infrastructure development. Digital twinning technology offers a solution to these challenges by amalgamating physical systems with numerical models. Digital twinning could realize real-time data interaction between physical systems and dynamic simulation and modeling by constructing a digital mirror image of actual systems, providing precise information and decision support for the operation and management of tangible systems. Integrating digital twinning into power grid development can revitalize the industry, as this technology enables the remote monitoring and operation of power grid equipment, improving grid efficiency and safety. Digital twinning has substantial potential to address grid challenges and provide vital support for the nation’s energy supply and economic progress [1,2,3].

The application of digital twinning technology in civil engineering has become a hot topic recently [4,5]. Digital twinning employs virtual entities to represent physical objects, allowing for the transfer of data from the physical to the virtual domain, thereby capturing the complete lifecycle processes of the physical objects [6]. Yoon et al. [7] proposed an innovative framework and approach for constructing digital twins throughout the lifecycle of buildings. Liu et al. [8] achieved real-time and accurate power forecasting for wind turbine units through digital modeling. Ritto et al. [9] created digital volumetric models for compromised structures and explored a range of damage scenarios utilizing discrete physics-based computational models. Xu et al. [10] proposed a digital twin health monitoring information model for buildings to simultaneously locate the building in space and monitor data. These researchers have significantly enhanced the comprehensive monitoring of the entire lifecycle processes of physical objects by developing digital twin models for structures or objects and conducting corresponding research. Vulnerability analyses of long-span transmission towers under various conditions such as wind fields and earthquakes have been conducted by Li et al. [11], Liu et al. [12], Zheng et al. [13] and Gong et al. [14]. They discussed different modes of structural failure under distinct conditions and explored the primary factors contributing to the deterioration of transmission tower structural performance. However, there is no standardized methodology for the application of engineering digital twins. Additionally, it is difficult to experiment with transmission towers because of their complexity and long testing periods. To address these challenges, the sandbox model reduces the complexity of the transmission tower testing procedures, while the CDEM method realizes a rapid calculation of the dynamic response of transmission towers under wind conditions, thereby reducing the testing duration. Kong et al. [15] and Wang et al. [16] monitored the deterioration of buildings and evaluated post-earthquake structural conditions by a photogrammetry-based digital twin framework. These researchers effectively demonstrated the strengths of digital twin methodologies in the field of engineering.

This paper presents an implementation approach to the entire lifecycle of a physical model under a digital twin technology framework. The framework is composed of the transmission tower sandbox model, camera monitoring equipment and digital simulation software. The sandbox model is as the physical object of study, the camera equipment captures the model’s geometric dimensions to create a virtual model and the digital simulation software can simulate the entire lifecycle process of the virtual model. This approach allows for the establishment of a high-precision continuous mathematical model, enabling a holographic mapping from the real space to the virtual digital space. The main content of this paper is as follows:
The 3D numerical model is established through the algorithmic processing of two-dimensional images and the creation of corresponding digital three-dimensional scenes. The accuracy of the model is verified by comparing the results from physical and digital models.Within the framework of CDEM, we simulated the mechanical characteristics of transmission towers under unidirectional and variable wind conditions. We analyzed the impact of loads on structural displacement and acceleration.We converted historical on-site wind speeds into loads, accounting for wind direction effects. We coupled multiple factors through coding, such as load, direction and time, for transmission towers. We confirmed the feasibility and high accuracy of the model, algorithms and simulation methods by comparing the experimental data with the numerical simulation results.

## 2. Numerical Calculation Method

### 2.1. Continuum-Discontinuum Element Method (CDEM)

The CDEM [17,18] method is an explicit dynamic numerical method that couples finite and discrete elements. The CDEM method establishes the control equations based on the Lagrangian energy system.

The equation is shown as [19,20]:(1)$$\frac{d}{dt}\left(\frac{\partial L}{\partial v_{i}}\right)-\frac{\partial L}{\partial u_{i}}={\;Q}_{i}$$
where $u_{i}$ and $v_{i}$ represent generalized coordinates, *L* represents the energy of the Lagrangian system and *Q*_i_ represents the work done by non-conservative forces. The governing equation of the CDEM method is expressed as [19,20]:(2)$$M\ddot{u}+C\dot{u}+Ku\;+{\;K}_{c}u_{c}+C_{c}{\dot{u}}_{c}=F$$
where $\ddot{u}$, $\dot{u}$, *u*, $u_{c}$ and ${\dot{u}}_{c}$ represent the acceleration vector, velocity vector, displacement vector, relative displacement vector of the virtual crack and relative velocity vector, respectively. M, C, K, $K_{c}$, $C_{c}$ and F represent the mass matrix, damping matrix, stiffness matrix, contact stiffness matrix, contact damping matrix and external load matrix of the unit.

The power transmission tower is a spatial truss structure, where some members are connected through welding and bolting to transfer forces (i.e., bending moments). In this work, the numerical model is reconstructed using the “pile” element. Each “pile” structural element is defined by its geometric form, material parameters and coupling-spring properties. It provides beam-like structural properties for determining the ultimate plastic moment capacity. Additionally, it can simulate frictional effects in both normal and shear directions.

Every “pile” element has its own coordinate system. This system is used to define cross-sectional moments of inertia and applied distributed loads. It also sets the conventions for force and moment distributions within each element, as illustrated in Figure 1. The local coordinate system is determined by two nodes, marked as 1 and 2 in Figure 1, and is described by vector Y with three rules: (1) the central axis aligns with the *x*-axis, (2) the *x*-axis points from node 1 to node 2, (3) the *y*-axis is aligned with the projection of Y onto the cross-sectional plane (i.e., the plane whose normal is directed along the *x*-axis). These members can undergo axial yielding, and the yield strength depends on the inherent strength of the materials.

### 2.2. The Transformation Relationship between Images and 3D Models

In three-dimensional image processing, two three-dimensional coordinate systems and two two-dimensional coordinate systems need to be considered [21]. The four coordinate systems are nested as follows: world → camera → image → pixels. Once the world coordinate system is determined, it remains unchanged. The camera coordinate system undergoes transformations relative to the world coordinate system. The image coordinate system remains stable relative to the camera coordinate system. Figure 2 illustrates the relationships among these coordinate systems.

In Figure 2, $O_{w}$, $X_{w}$, $Y_{w}$ and $Z_{w}$ represent the world coordinate system, where the origin is the real-world coordinate system of the camera. $O_{c}$, $X_{c}$, $Y_{c}$ and $Z_{c}$ represent the camera coordinate system, where the origin is located at the optical center of the camera. o, x and y represent the image coordinate system.

The transformation relationships between three coordinate systems can be described using the extrinsic matrix and the intrinsic matrix. The extrinsic matrix represents the position and orientation of the camera relative to the reference point, and it is used to describe the transformation relationship between the world coordinate system and the camera coordinate system. On the other hand, the intrinsic matrix represents the fixed parameters of the camera, specifically the internal parameters that project objects from the camera coordinate system to the image plane. The intrinsic matrix is commonly used to describe the relationship between the camera coordinate system and the image coordinate system. By performing coordinate system transformations, the relationship between the two-dimensional image coordinate system and the physical world coordinate system could be determined [22]:(3)$$Z_{c}\left[\begin{array}{c} u\\ v\\ 1 \end{array}\right]=\left[\begin{array}{ccc} \frac{1}{dx} & 0 & u_{0}\\ 0 & \frac{1}{dy} & v_{0}\\ 0 & 0 & 1 \end{array}\right]\left[\begin{array}{cccc} f & 0 & 0 & 0\\ 0 & f & 0 & 0\\ 0 & 0 & 1 & 0 \end{array}\right]\left[\begin{array}{cc} R & T\\ 0 & 1 \end{array}\right]\left[\begin{array}{c} X_{w}\\ Y_{w}\\ Z_{w}\\ 1 \end{array}\right]=\left[\begin{array}{cccc} f_{x} & 0 & u_{0} & 0\\ 0 & f_{y} & v_{0} & 0\\ 0 & 0 & 1 & 0 \end{array}\right]\left[\begin{array}{cc} R & T\\ 0 & 1 \end{array}\right]\left[\begin{array}{c} X_{w}\\ Y_{w}\\ Z_{w}\\ 1 \end{array}\right]$$

In this transformation relationship, the intrinsic matrix can be accurately obtained through camera calibration. The intrinsic matrix is assumed to remain unchanged when the camera application environment remains static. However, the extrinsic matrix changes when the camera position changes. In this study, we utilize coordinate transformations to handle captured image data and reconstruct a three-dimensional virtual model that corresponds to the real-world scenes captured in the two-dimensional images.

### 2.3. The Relationship between Wind Speed and Load Conversion

To simulate wind pressure in real-world scenarios, it is important to consider the dynamic characteristics of wind, in which both direction and magnitude change with time. In this paper, a wind direction simplification method is employed, as shown in Figure 3. In Figure 3a, the wind direction is represented in 16 different directions. The trigonometric values for each angle are then calculated to determine the load components on the *x*- and *y*-axes for the wind at each angle. These load components are subsequently applied to different nodes of the structure, allowing for an accurate simulation of the wind load on the transmission tower.

The wind load values for the tower are computed in accordance with the regulations specified for overhead transmission line loads. The formula for calculating the standard wind load on the tower is as follows [23]:(4)$$W_{s}=W_{0}\cdot \mu_{z}\cdot \mu_{s}\cdot \beta_{z}\cdot B_{2}\cdot A_{s}$$
where $W_{s}$ represents the standard value of the tower wind load (kN), $\mu_{s}$ is the component shape coefficient, $\beta_{z}$ is the tower wind vibration coefficient at height *z* and $B_{2}$ is the wind load increase coefficient for tower components. For conditions designed with ice, 1.1 is taken for a 5 mm ice zone, 1.2 for a 10 mm ice zone and 1.0 for no ice conditions. $A_{s}$ is the calculated value of the projected area of the components on the windward side.

The value of $W_{0}$ is calculated as [23]:(5)$$W_{0}={\;V}_{0}^{2}/1600$$
where $V_{0}$ is the basic wind speed (m/s), $W_{0}$ is the reference wind pressure (kN/m^2^) and $\mu_{z}$ is the wind pressure factor for the height variations of the transmission tower.

### 2.4. Wind Field Calculation

This study developed a method that can consider the multivariate effects of load, direction and time during the simulation process. The GDEM software used in the study did not provide complex load application functions, which limited the study of the mechanical response of transmission towers in different scenarios. To address this limitation, the specific numerical values of wind loads are computed and stored in a text (txt) file. Afterward, the algorithm reads and stores these values in arrays. These values are then applied to the nodes of the respective component groups. The position of each component in the model is determined based on its group identifier and the load values from the specific arrays are applied to the nodes of these components. This iterative process continues until all load values in the array are computed. Through this approach, the fluctuations in wind force magnitude and direction at various locations within the model could be considered. Furthermore, adaptations to load application timing and computation time were achieved by modifying the time step for calculations, with the specific steps outlined in Figure 4.

### 2.5. Method Validation

This paper compares the velocity curves of the physical model of the transmission tower with that obtained from the numerical model under operating conditions to validate the proposed method. The velocity curve of the physical model is monitored in real-time using sensor equipment. Sensor devices were used to measure and monitor various physical quantities and experimental parameters of the test structure. To ensure the validity of the experiments, the material parameters of the numerical model were consistent with the actual materials of the physical model, and the geometric dimensions were kept consistent with the physical model. Furthermore, the simulation monitoring points for the numerical model were positioned at the same locations as those on the physical model, as depicted in Figure 5.

The experimental monitoring results have a good agreement with the numerical simulation results. As shown in Figure 6, point A represents the monitoring point for the transmission tower experiment and point B represents the monitoring point for the digital simulation calculations. This confirms the suitability of the wind load conversion method and its application process for numerical simulation research. The error in peak velocities within each second is within 5%.

## 3. Image-Based Modeling

The image-based modeling method provides an efficacious means for the visual examination and surveillance of transmission tower structures by generating detailed 3D models from visual data. This approach assures safety and dependability while also resulting in substantial time and resource economization.

An image-based modeling approach was employed in this study to establish a digital model of the sandbox system, achieving the virtualization and digitization of the physical entity. The geometric shape and dimensions of the physical entity were transferred onto virtual objects through software calculations. This served as the basis for constructing a computable three-dimensional numerical model.

### 3.1. Digital Model

The sandbox system is a scaled-down mountain model constructed based on the contour lines of a mountain surface, with simplified treatment on the remaining areas. The model includes many features such as trees, transmission towers, solar panels and power plants. The transmission tower model was fabricated using 3D printing technology and is depicted in Figure 7a. The model is placed on a cubic table measuring 1.5 m × 1.5 m × 1.5 m.

A three-dimensional model is created by capturing a complete rotation of the object with a camera. A point cloud model is established by segmenting the video frames and then identifying and extracting key points from the images [24]. Colorization is applied afterward to generate the digital three-dimensional model, as illustrated in Figure 7b.

### 3.2. Transmission Tower Model and Materials

The digital model of the transmission tower was created based on the actual physical object. The geometric dimensions were measured to be 43 cm in height and printed using Acrylonitrile Butadiene Styrene (ABS) material. The physical and mechanical parameters of the model are listed in Table 1. The geometry and dimensions of the numerical model are consistent with those of the physical model. The physical object and numerical model are illustrated in Figure 8.

The transmission tower model is composed of 639 connected elements. The model has a height of 43 cm and a cross-arm length of 24 cm. To monitor the acceleration and displacement of nodes, four monitoring points are established. The model is divided into three groups from top to bottom. The distribution of the model and the monitoring points is illustrated in Figure 9.

## 4. Results and Discussion

### 4.1. Initial and Boundary Conditions

This paper analyzes the characteristics of transmission towers under variable-direction and unidirectional wind conditions by examining the acceleration and displacement at different locations of tower nodes.

Initial cases: In the calculation for variable-direction wind conditions, meteorological wind speed data published by the weather station are selected. According to the tower wind load standard values as outlined in Section 2.3, the wind speed is converted into nodal forces, which are applied as initial conditions for convenience. The load values are presented in Table 2. The three columns of data in the table represent the load values applied to structure groups 1, 2 and 3 in the x- and y-directions, respectively. During the wind field calculations, tower wind sway coefficients in different height segments are proportionally set to apply corresponding loads to the tower groups. Different load values are applied to the structures of different heights in the transmission tower.

In the case of unidirectional wind conditions, a set of wind force data at a 45° wind direction angle are established. To study the structural stability and mechanical response of the transmission tower under fixed wind forces, the simulation lasted for 15 s at multiples of 100%, 130%, 150% and 200% of the wind force magnitude. Monitoring points were positioned at the top of each tower and within each group, which were systematically numbered from the tower base to the top.

Boundary cases: The velocity at the bottom nodes of the transmission tower model is constrained to 0, thereby ensuring the stability of the bottom node positions.

### 4.2. Simulation of Strong Winds on Transmission Tower

Transmission towers, as a type of highly flexible structure, have vibration characteristics like simply supported beams. Therefore, for simplification purposes, transmission towers are treated as beam element models in calculations. In the CDEM, the members are modeled using the “pile” model, which considers the structural behavior of the beam, including the ability to specify the plastic moment capacity. Additionally, the axial force (normal force) between members is calculated, and the load-carrying effect at the top of the tower is evaluated. In this paper, the members are modeled with a “failure” model to assess and analyze the stress state of individual transmission towers.

Figure 10 shows the displacement–time history curves for the transmission tower under strong wind conditions. The x- and y-direction displacements at each monitoring point maintain a consistent pattern in terms of magnitude and temporal variation. Notably, the displacement at monitoring point 1 remains virtually unchanged at 0 mm. In contrast, the displacements for monitoring points 2, 3 and 4 exhibit distinct “step-like” increments. When the wind force doubles, the displacements at these monitoring points correspondingly increase by a factor of 2, indicating a straightforward linear relationship between displacement and wind load. It is worth noting that the time of increased displacement on the curves typically occupies around one-third of the wind force application process. During this stage, the single-tower structure stabilizes, and displacement ceases to vary.

The phenomenon known as whip-end effect occurs when a flexible structure is subjected to external excitations or loads, resulting in nonlinear vibration. Disturbance or excitation at one end of the structure leads to the propagation of vibration waves along the structure and the creation of reflections at the endpoint. These reflected waves interact with the propagating waves, which results in an increased amplitude and instability in the vibrations at the endpoint. As observed in Figure 10, the displacement peak values gradually increase with height. This indicates that the structure’s response to various forces becomes more pronounced as the height increases. At the endpoint of a flexible structure, a lower mass and higher inertia make it more sensitive to external excitations. When these excitations propagate to the endpoint, they generate reflected waves that interact with propagating waves, ultimately leading to increased amplitude and vibration instability. It is worth noting that previous literature has also addressed similar factors [25,26].

When designing and optimizing transmission towers, it is crucial to pay special attention to the characteristics of the top structure. Rational design and optimization measures can mitigate the adverse effects of the whip-end phenomenon on the structure, thereby enhancing its stability and vibration suppression capabilities.

As observed from Figure 11 and Figure 12, the displacements of the nodes on transmission tower increase with the augmentation of wind loads. The displacements initiate from the top of the tower and propagate downward.

Simultaneously, the displacements progressively decrease from the top to the bottom of the tower due to the lower tower components being loosely arranged and having a smaller windward area compared to the upper sections. The maximum displacement at the tower top corresponds to the peak values on the curves. The maximum displacement is 8 mm in the x-direction and 7 mm in the y-direction, resulting in a 1 mm difference between the two directions. This discrepancy is attributed to the varying structural configurations on the windward side of the transmission tower, where form factors differ, ultimately leading to greater displacements induced by the applied forces in the x-direction.

### 4.3. Simulation of Wind Fields on Transmission Towers

The mechanical characteristics of a transmission tower in a complex wind field were investigated by applying the wind loads from Table 2. The total calculation time was 49 s, with each set of data being applied for 1 s. The displacement–time history curves and acceleration–time history curves are depicted in Figure 13 and Figure 14, respectively.

As is shown in Figure 13, the displacements in the x- and y-directions exhibit a disordered behavior with no apparent regularity in the conditions of a complex wind field. By comparing the amplitudes in the x-, y- and z-directions, it can be observed that the z-direction displacement changes remain relatively stable and noticeably smaller than those in the x- and y-directions. The amplitude of displacement in the x-direction is three times greater than that in the y-direction, which is attributed to the tower structure’s design. The x-direction side encompasses a larger projected force-receiving area, causing it to bear a greater load under wind or external forces, consequently resulting in the observed displacement disparity. As the height of the monitoring points increases, the peak displacements also increases due to the oscillations induced by the wind affecting the tower structure. Consequently, the node displacement and acceleration generated by these effects at the top become larger. Additionally, it is essential to account for wind vibration coefficients and height variation factors for wind pressure when designing large and extra-large transmission tower structures. This is because the meteorological data collected daily primarily focus on wind speeds at the ground level. This consideration becomes crucial because structural elements at greater heights are subject to higher wind forces.

Figure 14 illustrates the acceleration–time history curves for the transmission tower subjected to complex wind field conditions, revealing intricate and diverse characteristics. It is worth noting that the acceleration magnitude increases as the height of the transmission tower increases. In the lower segments, the z-direction acceleration shows a significantly greater variation compared to the x- and y-direction accelerations. However, the acceleration data in the x-, y- and z-directions show relatively equal variations and magnitudes at a higher position. This indicates that the transmission tower structure at higher segments is influenced in all three directions under complex wind field conditions, while at lower segments, the z-direction exhibits more pronounced variations. These findings are consistent with previous research [27]. Consequently, in the lower segments of the structure, there should be a heightened focus on mitigating or eliminating potential risks associated with the self-weight of all elements in structural design and optimization. Conversely, for the higher segments, strategies such as reducing the windward projection area and increasing stiffness can be employed to minimize the impact of wind forces.

## 5. Conclusions

This paper presents an implementation approach to digital twins in a transmission tower sandbox model, considering the extensive application of digital twin technology in the engineering industry. The approach combines image recognition modeling, sensor monitoring and the CDEM. The feasibility of establishing a three-dimensional numerical model is investigated based on two-dimensional images and simulation studies under wind loading conditions.

Image-based modeling methods enable the rapid identification of geometric dimensions and material properties based on actual structures. This novel modeling method streamlines the modeling process for large-scale structures. However, the details of this method still require manual refinements. It indicates the potential for extensive optimization and further research in the future.

The main conclusions are as follows:The implementation approach of a digital twin system for transmission towers in a sandbox is proven to be effective and reliable. It involves the utilization of image recognition modeling, camera sensor equipment and the CDEM. The transfer of information data from the physical model to the virtual model is achieved through the camera equipment. The numerical models, created from processed two-dimensional images, exhibit geometric dimensions consistent with the physical model. The dimensions have a 5 mm error, demonstrating the accuracy of the system.The functionality developed in Java language successfully achieved the application of multi-variable loads on the structure, including load, time and direction. The load application algorithm was shown to be effective and accurate, with the simulation results demonstrating an error of less than 5% when compared to the experimental data.Under unidirectional wind conditions, the displacement of the transmission tower exhibits a distinct “step-like” variation, with a simple linear relationship between the displacement and load magnitude as the load magnitude increases. However, under variable-direction wind loads, the variations in amplitude peaks are caused by differences in projected force areas resulting from different structural forms. The maximum acceleration at each monitoring point is approximately twice that of its lower position. With increasing height, both the displacement and acceleration of the transmission tower also increase accordingly.

## Figures and Tables

**Figure 1 sensors-23-08657-f001:**
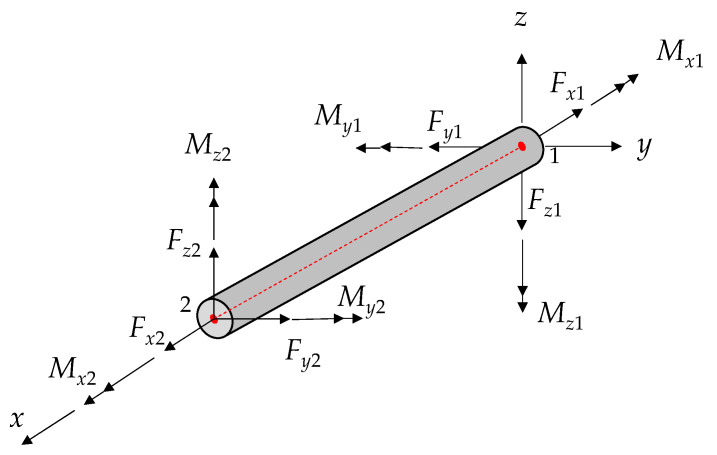
Distribution of forces and moments on the individual components.

**Figure 2 sensors-23-08657-f002:**
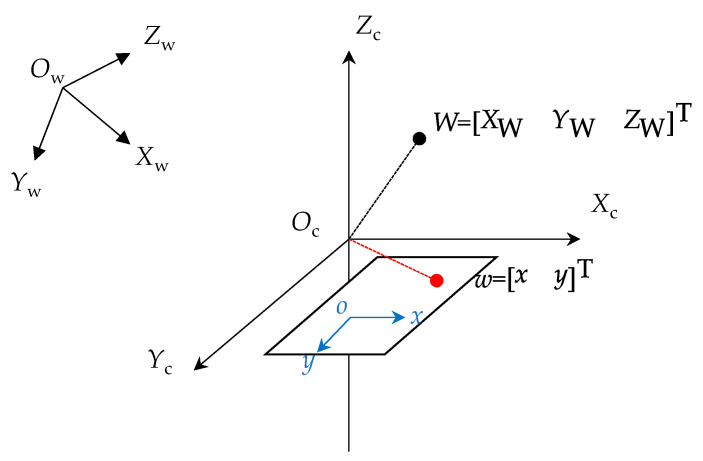
Relationship between two-dimensional and three-dimensional coordinate systems.

**Figure 3 sensors-23-08657-f003:**
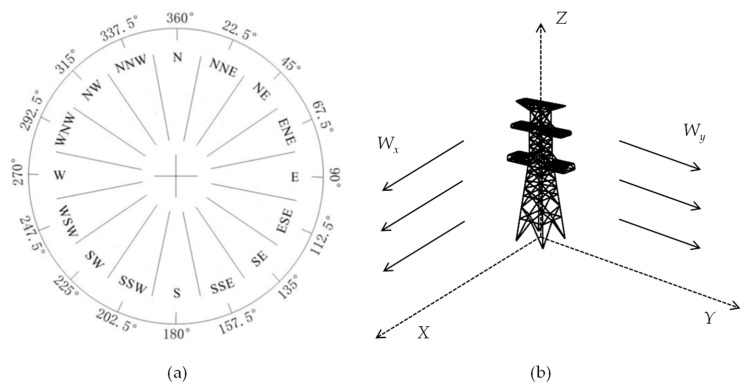
Wind direction decomposition (**a**) 16 wind directions; (**b**) direction of wind force application.

**Figure 4 sensors-23-08657-f004:**
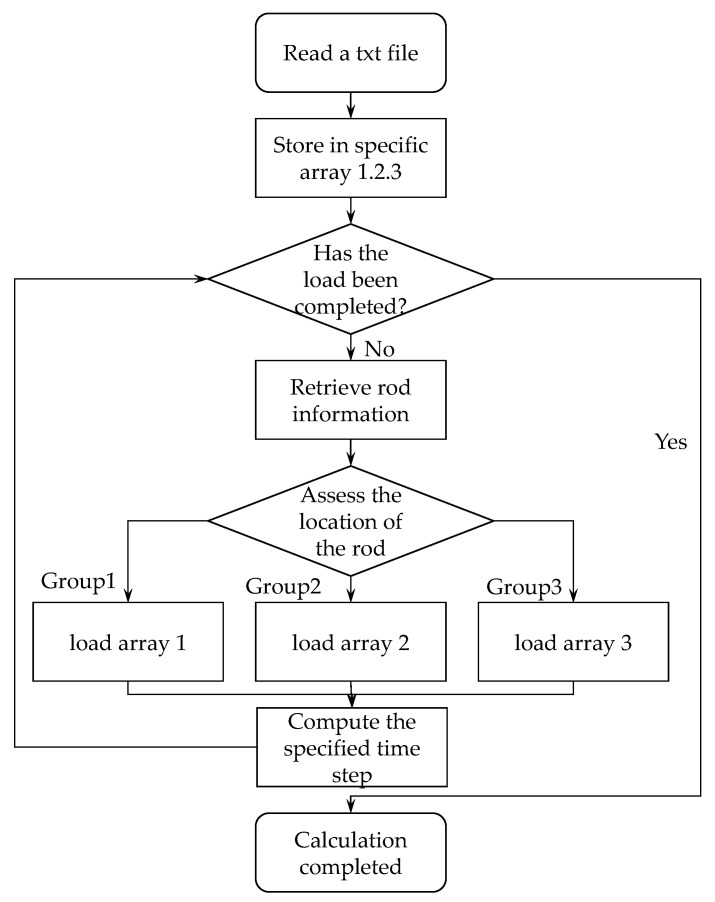
Wind load application procedure diagram.

**Figure 5 sensors-23-08657-f005:**
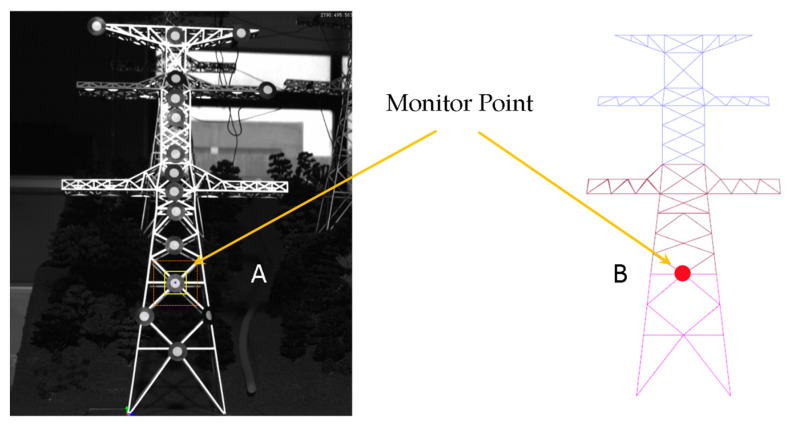
Positions of monitoring points.

**Figure 6 sensors-23-08657-f006:**
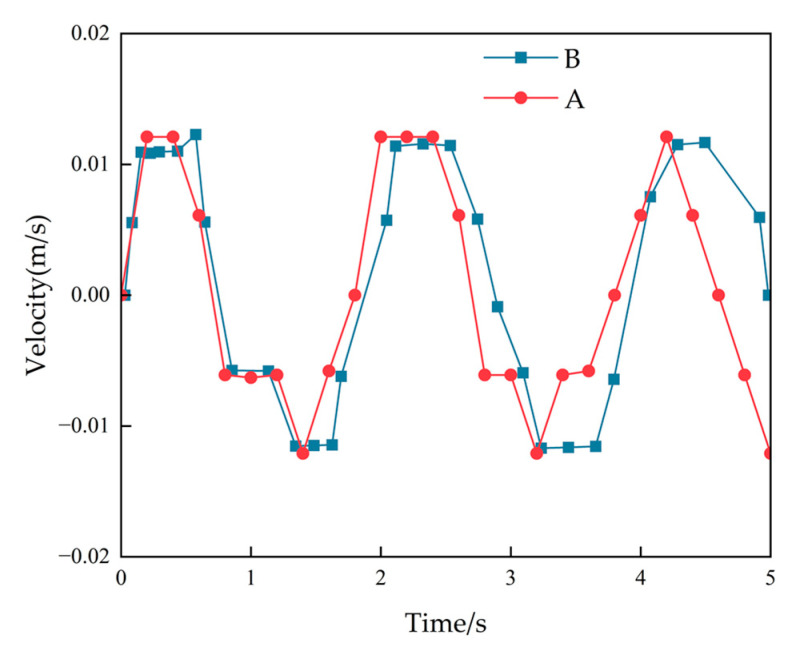
Comparative results.

**Figure 7 sensors-23-08657-f007:**
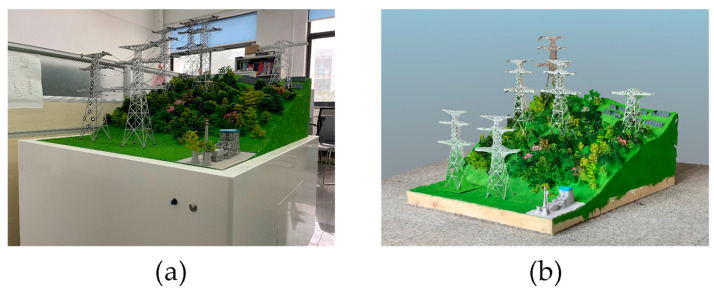
Sandbox model and digital model (**a**) sandbox model; (**b**) digital model.

**Figure 8 sensors-23-08657-f008:**
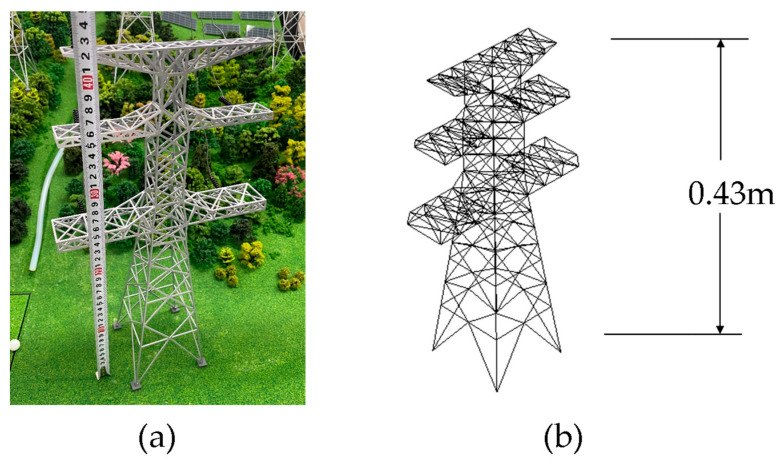
Physical and numerical models (**a**) physical model; (**b**) numerical model.

**Figure 9 sensors-23-08657-f009:**
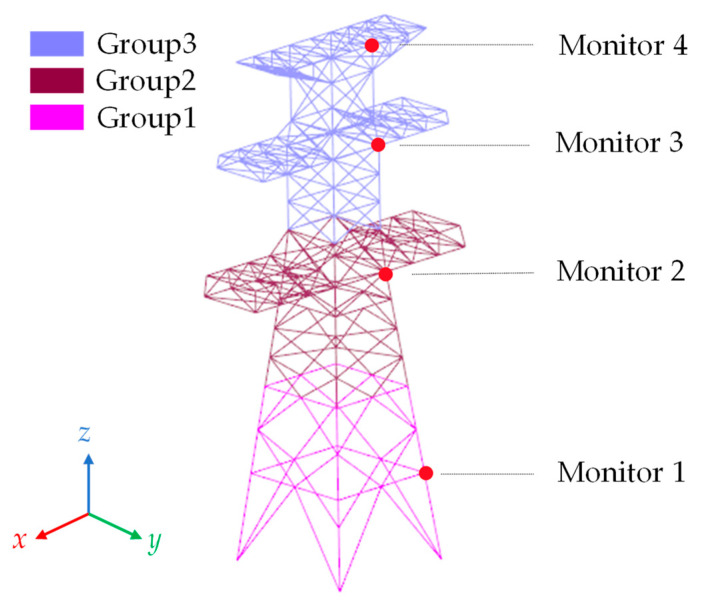
Grouping of transmission tower models and distribution of monitoring points.

**Figure 10 sensors-23-08657-f010:**
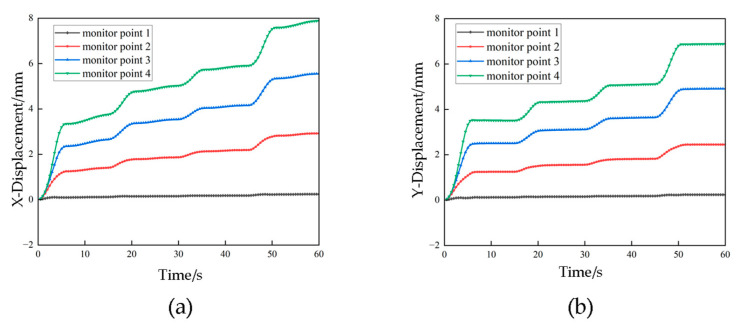
Displacement–time curve under strong wind conditions (**a**) X-direction displacement–time graph; (**b**) Y-direction displacement–time graph.

**Figure 11 sensors-23-08657-f011:**
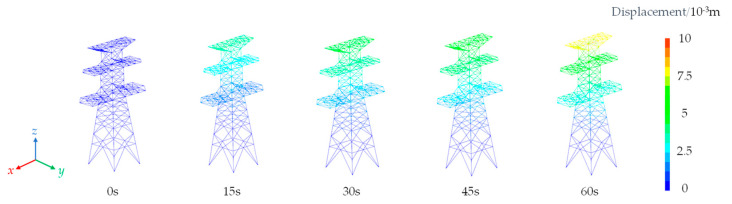
Total displacement in the x-direction of transmission tower.

**Figure 12 sensors-23-08657-f012:**
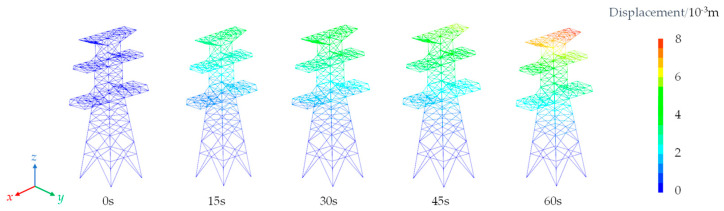
Total displacement in the y-direction of transmission tower.

**Figure 13 sensors-23-08657-f013:**
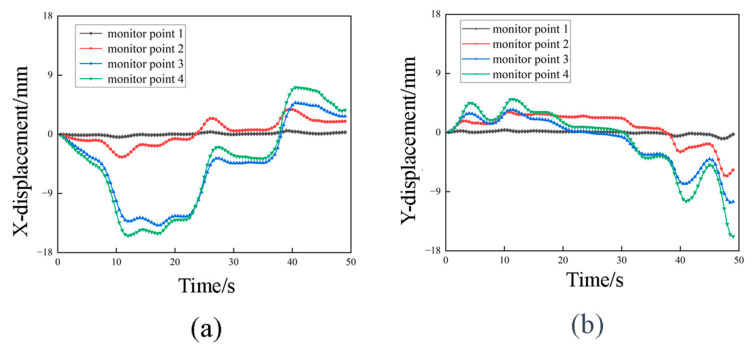
Displacement–time curve under complex wind field conditions (**a**) x-direction displacement graph; (**b**) y-direction displacement graph.

**Figure 14 sensors-23-08657-f014:**
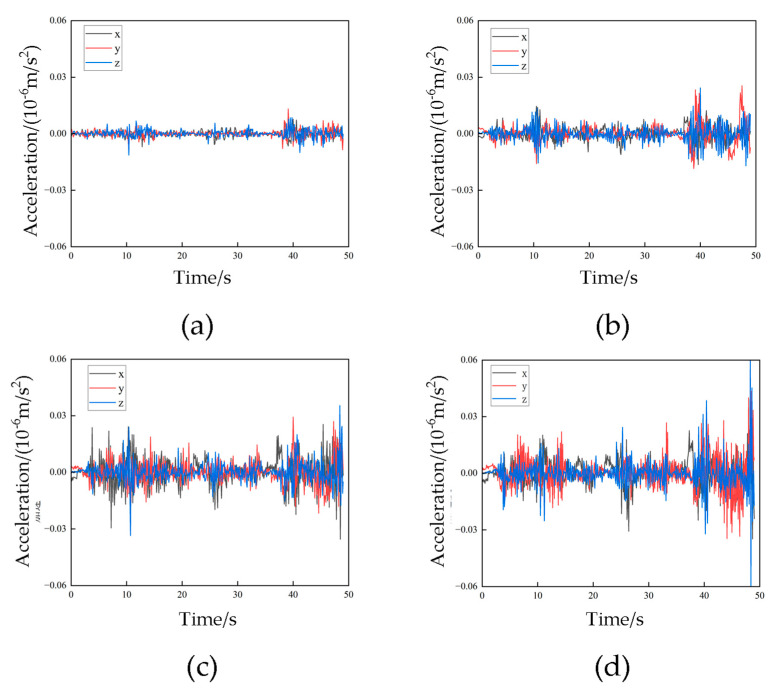
Acceleration–time curve under complex wind field conditions (**a**) Acceleration of monitoring point 1; (**b**) Acceleration of monitoring point 2; (**c**) Acceleration of monitoring point 3; (**d**) Acceleration of monitoring point 4.

**Table 1 sensors-23-08657-t001:** Physical mechanical parameters.

Density (kg/m^3^)	Young’s Modulus (MPa)	Poisson’s Ratio (-)	Tensile Strength (MPa)	Compressive Strength (MPa)
1050	2500	0.38	60	80

**Table 2 sensors-23-08657-t002:** Wind-force data.

Time/s	X-Direction/N			Y-Direction/N		
1	−4.506	−3.884	−11.590	10.725	9.535	7.031
2	0.000	0.000	0.000	11.607	10.319	7.610
3	−2.003	−1.726	−5.151	4.767	4.238	3.125
4	−2.003	−1.726	−5.151	−4.767	−4.238	−3.125
5	−4.832	−4.164	−12.427	−1.976	−1.757	−1.295
6	0.000	0.000	0.000	−5.159	−4.586	−3.382
7	−3.697	−3.186	−9.509	3.647	3.243	2.391
8	−8.318	−7.169	−21.395	8.206	7.296	5.380
9	−8.011	−6.904	−20.605	19.066	16.951	12.500
10	−8.011	−6.904	−20.605	19.066	16.951	12.500
11	−3.697	−3.186	−9.509	3.647	3.243	2.391
12	0.924	0.797	2.377	−0.912	−0.811	−0.598
13	0.924	0.797	2.377	0.912	0.811	0.598
14	−0.501	−0.432	−1.288	1.192	1.059	0.781
15	−4.506	−3.884	−11.590	10.725	9.535	7.031
16	−2.003	−1.726	−5.151	4.767	4.238	3.125
17	−2.003	−1.726	−5.151	4.767	4.238	3.125
18	5.229	4.507	13.450	0.000	0.000	0.000
19	0.924	0.797	2.377	0.912	0.811	0.598
20	−0.501	−0.432	−1.288	1.192	1.059	0.781
21	−0.924	−0.797	−2.377	0.912	0.811	0.598
22	0.000	0.000	0.000	5.159	4.586	3.382
23	8.318	7.169	21.395	8.206	7.296	5.380
24	5.229	4.507	13.450	0.000	0.000	0.000
25	11.765	10.140	30.261	0.000	0.000	0.000
26	1.307	1.127	3.362	0.000	0.000	0.000
27	−0.501	−0.432	−1.288	1.192	1.059	0.781
28	−0.924	−0.797	−2.377	0.912	0.811	0.598
29	0.501	0.432	1.288	1.192	1.059	0.781
30	0.924	0.797	2.377	−0.912	−0.811	−0.598
31	0.000	0.000	0.000	−11.607	−10.319	−7.610
32	2.003	1.726	5.151	−4.767	−4.238	−3.125
33	0.501	0.432	1.288	−1.192	−1.059	−0.781
34	0.000	0.000	0.000	−1.290	−1.147	−0.846
35	0.000	0.000	0.000	0.000	0.000	0.000
36	4.832	4.164	12.427	1.976	1.757	1.295
37	4.506	3.884	11.590	−10.725	−9.535	−7.031
38	19.326	16.657	49.709	−7.903	−7.026	−5.181
39	0.000	0.000	0.000	−46.428	−41.277	−30.439
40	5.229	4.507	13.450	0.000	0.000	0.000
41	−0.501	−0.432	−1.288	1.192	1.059	0.781
42	1.208	1.041	3.107	0.494	0.439	0.324
43	0.000	0.000	0.000	5.159	4.586	3.382
44	0.924	0.797	2.377	−0.912	−0.811	−0.598
45	−4.506	−3.884	−11.590	−10.725	−9.535	−7.031
46	0.000	0.000	0.000	−46.428	−41.277	−30.439
47	0.000	0.000	0.000	−46.428	−41.277	−30.439
48	0.000	0.000	0.000	−5.159	−4.586	−3.382
49	4.832	4.164	12.427	1.976	1.757	1.295

## Data Availability

Data available on request from the authors.

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
