# Peer review of "Wind Field Digital Twins Sandbox System for Transmission Towers"

_sensors, 2023, doi:10.3390/s23218657_

Round 1

Reviewer 1 Report

This paper is relatively simple and can be used for engineering guidance, but it is not suitable for scientific research.

English very difficult to understand/incomprehensible

Author Response

For the convenience of reviewers, I have arranged the revised replies and contents in the attachment and highlighted them in the paper.Please see the attachment.

Reviewer 2 Report

This study reveals solid work, but there are some things to be fixed, as follows:

A.      Remarks on the scientific content

If eq. (3) is not original, its source must be mentioned in text.

Table 2 contains some identical (sometimes consecutive) rows . Duplicates should be removed and the number of rows after this should be used in the line 327 instead of the number 49 (in this line Table 1 is actually Table 2!) Figures 11 and 12 should also be recomputed as they also address duplicate values from Table 2.

B. Typos

fractureable (lines 86/87)

2-nd word from line 269 is wrong

Many missing spaces (e.g. before brackets CDEM[17-18] line 86) a.s.o/

dots between Figure and no. of figure (e.g. line 168) should be removed in figure captions 

One should not put comma before  "etc."

Author Response

(The authors gave the same response as above.)

Reviewer 3 Report

This study introduces a digital twin system tailored to transmission towers, employing an innovative sandbox approach and harnessing imaging devices to capture the intricate physical details of these structures.

1. The research problem/gap is not clearly mentioned in the abstract. 

2. Most of the equations used in this research did not use references.

3. What are the technical contributions of this research? 

4. The simulation verifies the proposed contribution. Auhtor must use the real data to verify the proposed scheme. 

5. Results are not compared with related research.

 Minor editing of the English language required

Author Response

(The authors gave the same response as above.)

Round 2

Reviewer 1 Report

1. In Abstract, consider rephrasing "digitalization trends" to "digitalization trends within the field of engineering."

2. When introducing the "image processing modeling method," a brief explanation of the advantages and applicability of this method would be beneficial to help readers understand why this approach was chosen.

3. Confirm the format of variables and letters in formulas and pictures in the text.

4. Please check the units in Table 1.

5. In Conclusions: In the first point, specify the units for the error (e.g., 5 mm error) for clarity. Ensure consistency in terminology; for example, use "digital twin system" consistently instead of "engineering digital twin approach."

6. In the conclusion, it is advisable to provide suggestions for future research directions or further improvements to encourage readers to explore this field more deeply.

Minor editing of English language required

Reviewer 2 Report

I have no other comments to make

Reviewer 3 Report

Thanks for your work.
